# Targeting Cancer with CRISPR/Cas9-Based Therapy

**DOI:** 10.3390/ijms23010573

**Published:** 2022-01-05

**Authors:** Katarzyna Balon, Adam Sheriff, Joanna Jacków, Łukasz Łaczmański

**Affiliations:** 1Laboratory of Genomics & Bioinformatics, Hirszfeld Institute of Immunology and Experimental Therapy, Polish Academy of Sciences, 53-114 Wroclaw, Poland; lukasz.laczmanski@hirszfeld.pl; 2Faculty of Life Sciences and Medicine, Guy’s Campus, King’s College London, London SE1 9RT, UK; adam.sheriff@kcl.ac.uk; 3St. John’s Institute of Dermatology, Guy’s Campus, King’s College London, London SE1 9RT, UK; joanna.jackow@kcl.ac.uk

**Keywords:** cancer, genetic therapy, CRISPR/Cas9, targeting

## Abstract

Cancer is a devastating condition characterised by the uncontrolled division of cells with many forms remaining resistant to current treatment. A hallmark of cancer is the gradual accumulation of somatic mutations which drive tumorigenesis in cancerous cells, creating a mutation landscape distinctive to a cancer type, an individual patient or even a single tumour lesion. Gene editing with CRISPR/Cas9-based tools now enables the precise and permanent targeting of mutations and offers an opportunity to harness this technology to target oncogenic mutations. However, the development of safe and effective gene editing therapies for cancer relies on careful design to spare normal cells and avoid introducing other mutations. This article aims to describe recent advancements in cancer-selective treatments based on the CRISPR/Cas9 system, especially focusing on strategies for targeted delivery of the CRISPR/Cas9 machinery to affected cells, controlling Cas9 expression in tissues of interest and disrupting cancer-specific genes to result in selective death of malignant cells.

## 1. Introduction

Cancer is responsible for 20 percent of deaths in Europe, making it the third largest cause of mortality [1,2]. This life-threatening disease is caused by the step-wise accumulation of mutations and epigenetic changes in cellular genome, leading to sustained proliferation, resistance to cell death signals and growth suppressors, as well as increasing genetic instability during the process of tumorigenesis. Important features necessary for cancer progression also include angiogenesis, invasiveness (metastatic potential), pro-inflammatory activity and evasion of the immune system [3,4]. Mutations granting proliferative advantage to cells undergoing tumorigenesis, for example, activating oncogenes or deactivating suppressor genes, are termed “driver mutations”. “Passenger mutations”, on the other hand, are all of the accompanying gene alterations observed in cancer cells due to their genetic instability [5]. The plethora of possible genetic modifications occurring during the cancer formation and the randomness of these mutations create a genetic barcode specific to the patient (inter-tumour heterogeneity) or even to a single cancerous lesion (intra-tumour heterogeneity) [6]. It is worth noting that some mutations are distinctive to certain types and sub-types of cancer, providing bases for either universal or tailored therapies, compatible with the molecular profile of neoplastic cells.

Conventional cancer therapies such as the surgical removal of solid tumours, radiotherapy and chemotherapy, can suffer from low specificity and are encumbered with multiple and severe side effects. Therefore, current trends in oncology focus on increasing the selectivity and safety of developed therapies, with the rise of hormone therapy, immunotherapy and targeted drugs, including anti-angiogenics and growth blockers [7,8]. The recent discovery of precise gene editing tools has heralded an era in which modern treatment options can include therapies which directly target genes responsible for the continuous growth and survival of cancerous cells.

In principle, gene editing techniques employ various nucleases to introduce single or Double-strand breaks (DSBs) to the DNA strand at a specific site to install a permanent change in the genome. The first attempts at genetic therapies in oncology were based on Zinc-finger nucleases (ZFNs) or Transcription activator-like effector nucleases (TALENs) [9]. Both systems employ nuclease FokI and recognise targeted sequence with carefully engineered protein sets which physically interact with selected DNA fragment. Unfortunately, therapies based on ZFNs or TALENs are less feasible for diverse applications as the process of re-engineering guiding proteins is costly and time-consuming [9,10].

Seminal work in 2012 developed a gene editing tool from the adaptive immune system of bacteria and archea which has proved simpler, more affordable and reliable compared to previous gene editing tools [11,12,13,14]. The clustered regularly-interspaced short palindromic repeats (CRISPR)/CRISPR- associated protein 9 (Cas9) system uses short complementary strands of RNA, known as single guide RNA (sgRNA), to recognize and bind to specific DNA sequences through Watson-Crick base pairing. This directs a Cas9 nuclease to perform site-specific DNA cleavage mediated by its two nickase domains, RuvC and HNH. In addition to being easier to program compared to ZFNs and TALENs, CRISPR/Cas9 facilitates ‘multiplexing’ of more than one sgRNA to target multiple DNA sites simultaneously.

A potential limitation with CRISPR/Cas9 technology is the requirement of a Protospacer adjacent motif (PAM) site on the DNA upstream of the target site to enable the sgRNA to bind [15]. The PAM site recognized by the most common Cas9 isolated from *Streptococcus pyogenes*, SpCas9, is the trinucleotide NGG (in the 5’ to 3’ direction), where N is any of the four DNA nucleotides. Fortunately, this PAM site recurs in the human genome every 8 base pairs and Cas9 nucleases with other PAM requirements have also been described or created [16,17,18]. The DSBs created by CRISPR/Cas9 are repaired through two endogenous cellular mechanisms: Non-Homologous End Joining (NHEJ) or Homology Directed Repair (HDR) [19,20]. NHEJ is an error-prone process which reattaches cut ends of DNA but can also cause the random insertion or deletion (indels) of nucleotides at the break site, possibly resulting in complete gene knock-out, if a frame-shift is induced in a coding region. HDR processes, by contrast, enable the precise alteration of a sequence following a DSB by incorporating a DNA donor template containing an edit. However, HDR is inefficient and therefore can limit the percentage of cancer cells which undergo high fidelity editing [21]. Furthermore, as the NHEJ process predominates, the stochastic indel formation at the break site can possibly cause unpredictable changes in gene function. To surmount this, proprietary small molecule reagents have been developed and show promise in inhibiting NHEJ and enhancing HDR rates by up to 7-fold [22].

However, the most significant drawback with using CRISPR/Cas9 is its propensity to cleave regions of the genome with a similar DNA sequence to the target site, a phenomenon known as ‘off-target editing’. Off-target editing can introduce DSBs and indel mutations in critical regions causing deleterious changes in gene function or even tumorgenicity itself. Harnessing in silico prediction tools can help counter this by designing sgRNAs less likely to exhibit off-target editing [23]. Modification to the Cas9 nuclease has demonstrated some benefits in improving the safety of CRISPR/Cas9 therapy, (see Table 1). Highly specific Cas9 variants have been shown to reduce off-target editing to almost undetectable levels [24] and techniques which employ two evolved Cas9 ‘nickases’, which each instituting a single-stranded break on the DNA, have also demonstrated a reduction in indel formation and off-target editing [25]. Rigorous genome-wide screening for off-target activity post-editing would be a cornerstone of any work on cancer therapy.

A large proportion of genetic mutations in cancer is comprised of single-nucleotide changes in genes like NOTCH [26,27], EGFR [28] and KRAS [29]. These point mutations are potentially targetable by CRISPR-related RNA-programmable deaminases, known as base editors (BEs) (see Table 1). BEs unwind a short region of DNA known as the ‘editing window’ at the specific site and uses a deaminase to enzymatically alter the base [30]. A significant advantage of base editing is that it obviates the introduction of DSBs and therefore prevents activation of the NHEJ pathway and indel mutations, unlike conventional CRISPR/Cas9. Furthermore, BEs do not rely on HDR repair so higher editing efficiencies can be pursued, and demonstrated, and a DNA donor template is not required which simplifies mRNA-based delivery systems [31]. There are currently two main groups of BEs, Adenine base editors (ABEs) which convert A-T base pairs to G-C base pairs and Cystosine base editors (CBEs) which engender the reverse change of C-G base pairs to T-A base pairs.

On the other hand, base editors also exhibit off-targeting activity like conventional CRISPR/Cas9 editors and there is an additional risk of the deamination of other ‘bystander’ nucleotides within the same editing window. A further limitation of base editing is that it is constrained by its mechanism to only effecting transition point mutations [32] and therefore different tools are required to introduce other mutations in the genome.

The recent expansion of the gene editing toolbox with ‘prime editing’ may resolve this issue [33]. Prime editing is a novel DSB-free method of manipulating the genome with targeted insertions, deletions, and all base substitutions, conferring greater flexibility compared to BEs. Prime editing guide RNAs (pegRNAs) are deployed alongside a Prime editor (PE) protein which creates a single-stranded break at the target site and drives the inclusion of a new editing sequence through reverse transcription. The latest PE variants demonstrate comparable editing efficiencies to CRISPR/Cas9 and base editing tools, with diminished off-target and bystander editing [34], suggesting future promise for targeting oncogenic mutations.

**Table 1 ijms-23-00573-t001:** Available modifications of Cas9 protein employed to alternatively affect recognised DNA sequence, without introduction of DSBs [10,23,35,36,37,38,39].

Cas9 Modification	Application
Cas9 nickase	Creates a single-stranded DNA break. Pairing two nickases to target both DNA strands reduces off-target activity and indels introduction
deadCas9 (dCas9)	Harbours no nuclease activity, serves as a tool for delivering other active compounds and proteins to selected DNA sequence
dCas9 with KRAB domain	Inhibits expression of downstream gene
dCas9 with VP64	Activates expression of downstream gene
dCas9 with methylase/demethylase or histone deacethylase	Provides control over epigenetic regulation
Cas9 nickase with adenine or cytosine deaminase (Base editors; BEs)	Enables transition substitutions (A>G, T>C, G>A or C>T) with high efficiency and no double-strand DNA breaks
Cas9 nickase with a reverse transcription domain (Prime editors; PEs)	Facilitates introduction of any substitution, insertion or deletion mutation with high efficiency and no double-strand DNA breaks

A unique problem of gene editing therapies targeting cancerous cells is that the driver mutations occur usually in genes responsible for proper functioning of cells and these mutations are present only in a portion of patient’s cells derived from healthy and normal tissue. Therefore, systemic therapy-without the use of cancer-targeting techniques-could lead to dangerous and unpredictable consequences, including death of affected non-cancerous cells throughout the organism. This review aims at presenting the most recently explored strategies for cancer-specific CRISPR/Cas9-based oncological therapies, that do not affect healthy cells and tissues.

## 2. Targeted Delivery

The CRISPR/Cas9 system, before it can introduce programmed genetic changes, needs to be delivered to the cells of interest and penetrate the barriers of the cell and nuclear membranes. In vitro delivery thus far has been mostly based on simple and efficient non-specific techniques introducing plasmids encoding Cas9 and sgRNA, such as electroporation or microinjection [40]. Plasmid-based transfection in vivo is possible e.g., in mice via hydrodynamic infusion through the tail vein, but suffers from low efficiency. Moreover, this technique is not suitable for use in humans [18,37,41]. To ensure safety and effectiveness of therapy designed for patients, CRISPR/Cas9 needs to be first enveloped in a delivery capsule, either viral or non-viral such as vesicles. The CRISPR/Cas9 tools can be transported as a ready-assembled protein-sgRNA complex or form the complex intracellularly. It can also be delivered as mRNA or as genes encoding Cas9 and the sgRNA which require subsequent expression [42,43]. Each of these delivery modes has its relative advantages and limitations and may be favourable in certain experimental setting (see Table 2). These basic delivery models need to be further properly adjusted and modified to ensure selective uptake of carriers containing CRISPR/Cas9 by cancerous cells. Herein we describe some of the recently explored ideas and strategies for targeted delivery of CRISPR/Cas9 system.

### 2.1. Adeno-Associated Virus Serotypes

Initially viral vectors were used as delivery model for genes, since they have a natural ability to introduce their own genetic material to infected cells. However, some concerns were raised about their safety due to immunogenicity and inherent pathogenicity. A solution was found in a better tolerated, non-pathogenic and non-integrating recombinant form of adenovirus preparations’ contaminant named Adeno-associated virus (AAV) [45]. Several serotypes of AAV have been described and some display a particular affinity to different tissues and organs. In an experiment conducted on mice transfected with nine different serotypes of AAV via tail vein injection, AAV pseudotype 4 led to the highest expression of genetic material in lungs and kidneys followed by heart, while AAV 6 showed tropism to heart, brain and liver [46]. Unfortunately, the expression of the delivered luciferase gene was diminished but detectable in other tissues, meaning that genetic manipulations could affect multiple organs or even a whole body. While this may be acceptable or even desirable in therapies for congenital hereditary disorders, it is certainly not specific enough for cancer therapy. However, coupling AAV delivery with some of the targeted expression strategies described below, could result in precise and safe gene engineering.

Another obstacle to AAV-based CRISPR/Cas9 delivery is its low capacity. AAVs can pack up to approximately 5 kbp of DNA in a payload (including viral flanking sequences), while the SpCas9 gene itself is about 4.1 kbp long [9,18,45] and introduced genes usually require the addition of a suitable promoter to undergo transcription. The main strategy to circumvent this is to load the Cas9 gene and sgRNA into separate AAV vectors introduced simultaneously. However, this approach hinders the efficacy of delivery and gene manipulation, since each cell needs to be transfected with both versions of AAV. As the effectiveness of delivery remaines below 100%, the probability of successful outcomes decreases with each additional vector required to obtain desired results [47]. Alternatively, Cas9 orthologs encoded by shorter genes have been found in different bacterial species which can replace the SpCas9 in AAV payloads. The sequence of *Staphylococcus aureus* Cas9 (SaCas9) gene is 1 kbp smaller in size than Cas9 found in *Streptococcus pyogenes* and *Campylobacter jejuni* Cas9 protein (CjCas9) gene is shorter by 2.95 kbp than SpCas9 [48,49,50]. When considering the use of SpCas9 ortholog it is worth remembering, that alternative Cas9 proteins require different PAM sequences for successful gene manipulation.

### 2.2. Oncolytic Viruses

Another approach at viral vector delivery is based on Oncolytic viruses (OV) which are viruses that preferably replicate in cancerous cells. OV take advantage of the impaired response to pro-apoptotic signals in cancerous cells and the acquired resistance to apoptosis which protects normal, healthy cells from infection. Currently most frequently explored OV for oncological therapies include Herpes simplex virus (HSV), mumps, adenovirus, vaccinia virus and retrovirus [51,52]. Modifications introduced to these viruses aim at increasing their specificity, safety and immunogenicity. The immunostimulatory activity of OV is of utmost importance, restoring the natural anti-tumour response of patient’s immune system. Numerous clinical trials are therefore testing therapies based solely on oncolytic viruses as an anti-cancer agent [52,53], however, the propensity of OV to infect neoplasms, makes it a perfect candidate for a delivery system for CRISPR/Cas9. Recently, the myxoma virus (MYXV) was used as a vector for CRISPR/Cas9 targeting the NRAS gene in xenografts of embryonic rhabdomyosarcoma in mice after traditional viral vectors had failed. The large packaging capacity of the OV (161.8 kbp) augmented the utility of the oncolytic myxoma virus and allowed for the simultaneous targeting of multiple sequences. Replication of the administered MYXV was restricted to cancerous tissue and the CRISPR/Cas9-mediated NRAS inactivation resulted in reduction of tumour growth and increased survival of mice. However, eventually, all tumours relapsed [54].

### 2.3. Antibody- or Aptamer-Guided Lipid-Based Vectors

CRISPR/Cas9 delivery in non-viral lipid-based vectors is generally safer than viruses, but also less efficient [44]. Liposomal transporters offer a broad variety of available modifications, allowing for precise control over the physico-chemical properties of their surface and the introduction of targeting ligands [55]. Antibody-guided system emerged in combination with Lipid nanoparticles (LNPs), that comprise a well-known delivery system for genetic material, initially developed and standardized for treatment with small interfering RNA (siRNA) [49,56]. Recently, LNPs were used in a proof-of-concept study to target disseminated ovarian cancer in mice with CRISPR/Cas9. Complexes were constructed from ionizable cationic amino lipids coated with anti-EGFR antibodies, providing cancer-specific uptake. The vector contained mRNA for Cas9 and sgRNA targeting the PLK1 gene. Systemic intraperitoneal injection of LNPs resulted in their accumulation in EGFR-overexpressing cancer cells and resulted in an approximately 82% editing rate of PLK1 and increased survival by around 80% [57]. The elegance and usefulness of this solution lies in the antibody-guided neoplasm-specific delivery, since anti-EGFR can be easily swapped for another immunoglobulin targeting different antigen expressed on cells of interest.

A noteworthy protein-free alternative to antibodies was found in three-dimensional structures of single-stranded nucleic acids, termed aptamers. After *in vitro* selection, these molecules can specifically target and recognise certain cells or membrane proteins and have currently been employed for selective delivery of various drugs and treatments, including CRISPR/Cas9 [58,59]. In one study, ssDNA aptamers specific to osteosarcoma cells were used to create aptamer-guided lipopolymers containing plasmids with Cas9 and sgRNA targeting the VEGFA gene [60]. Another study used RNA aptamers recognising Prostate-specific membrane antigen (PSMA) expressed on prostate cancer cells. Vesicles guided by these aptamers were formed from cationic liposomes containing Cas9 and sgRNA targeting PLK1 [61]. Systemic injection of said ’aptamer-liposomes’ in mice burdened with either prostate cancer or metastatic osteosarcoma led to selective expression of CRISPR/Cas9 system in cancer cells and a suppression of tumour growth.

There has been recently an attempt to control release of CRISPR/Cas9 ribonucleoprotein from liposomes using light sensitive verteporfin. This proposed light-switch would allow for spatial selectivity of gene therapy for solid tumours via optic fiber insertion into the cancer tissue. However, both safety and efficacy of this method needs to be further improved [62]. Another approach at spatial control of CRISPR/Cas9 release was employed by Deng et al. in combinatorial anti-tumour therapy. Researchers used near-infrared radiation to mediate release of RNPs from endosomes and induce oxidative stress in exposed cells. Additionally, co-delivered chlorine e6 served as a photosensitizer and CRISPR/Cas9 targeted the gene encoding the NRF2 protein involved in antioxidative response. Altogether, these mechanisms led to the selective apoptosis of irradiated cells due to oxidative stress and cell damage caused by reactive oxygen species [63].

### 2.4. Extracellular Vesicles

A novel strategy for the possibly safer delivery of gene engineering tools is based on bio-compatible vectors mimicking natural cellular membranes. Extracellular vesicles (EV), either exosomes or microvesicles, are part of a communication system between cells and tissues, which distribute signals and molecules from the inside of one cell, including small fragments of genetic material. The composition of their outer layer is akin to the parent cell, granting them propensity to target certain cell types and their natural origin ensures low immunogenicity and good tolerability, unmatched by viral vectors or cationic liposomes [64,65]. The selectivity of EV can be further improved by introducing guiding molecules (aptamers or antibodies) to the surface of parenting cells which are transferred to newly created microvesicles. This method was used to successfully coat EV with CARs (Chimeric antigen receptors) targeting CD19+ B-cell malignancies [66]. Furthermore, exosome-AAV or exosome-liposome hybrids offer safe delivery method resistant to neutralizing anti-AAV antibodies and not burdened by inherent toxicity of liposomes [67].

## 3. Targeted Expression

### 3.1. Promoters Specific to Certain Type of Cancer Cells (AND Gate)

Cancer-specific promoters controlling genes overexpressed in tumour cells, such as hTERT, Ran, Brms1 or Mcm5 can be used in CRISPR/Cas9 gene therapy to achieve selective expression of Cas9 protein and/or the sgRNA targeting genes necessary for cell survival [68,69,70]. However, these promoters in healthy tissues exert lower, yet still observable activity, which is sufficient to affect the specificity of CRISPR/Cas9-based therapy and result in CRISPR/Cas9 ribonucleoprotein assembly in non-cancerous tissues. Significantly improved selectivity of expression was achieved through construction of a ’logical AND gate gene circuit’ with two promoters controlling one output gene. The first promoter, the bladder-specific hUPII, regulates transcription of the Cas9 gene. The Cas9 protein however exerts no activity, unless bound with the sgRNA, whose expression is controlled in this model by the cancer-specific hTERT promoter. The complete ribonucleoprotein complex exists only in cells fulfilling both conditions, i.e. are of bladder and cancer origin. The assembled CRISPR/Cas9 then targets the LacI gene encoding a suppressor protein, which inhibits expression of effector genes under the control of lac operator, and thereby CRISPR/Cas9 activates effector gene expression. This study used the output genes hBAX, p21 and E-cadherin. Their activation led to, respectively: apoptosis, growth arrest and decreased motility in bladder cancer cells [71]. Employing different tissue-specific promoters and/or providing alternative effector genes allows for targeting of virtually any tumour with high specificity.

### 3.2. Expression Dependent on Telomerase Activity

Telomerase is an enzyme which in developed organisms is active mostly in highly proliferating cells. It adds repetitive telomere sequences to the ends of chromosomes, protecting them from shortening and subsequent loss of important genetic information with each cell division. Apart from malignant tumours only some stem or germ-line cells harbour telomerase activity in adult humans [72,73,74,75]. Recently Dai et al. appropriated this relative specificity to develop a CRISPR/Cas9 system dependent on telomerase activity. This method requires delivering two sets of genes encoding firstly, dCas9-VP64 and sgRNA targeting telomeres under the control of constantly active promoters and secondly, nuclease Cas9 with a small DNA fragment attached, which is recognised by telomerase. In tissues distinguished by high telomerase activity the fragment preceding the Cas9 gene is extended with telomeres. Their sequence is recognised by the sgRNA guiding dCas9-VP64 protein, promoting the expression of the nearby downstream gene - in this case Cas9. Then, the Cas9 harbouring nuclease activity uses the same sgRNAs complementary to the telomeres to target them, leading to their shortening and degradation in all chromosomes. Consequently, cells undergoing further divisions die due to damaging and loss of the vital genetic information. The specificity of this system to cancer cells was demonstrated both *in vitro* in cell cultures and *in vivo* in mice xenografted with mixture of Hepa1-6 cells. Moreover, tumour growth in animals treated with this CRISPR/Cas9 system was significantly reduced. Interestingly, cultures of non-cancerous cells possessing naturally high telomerase activity were unaffected by this treatment [76].

### 3.3. CRISPR/Cas9 Switch Responsive to Micro-RNA

Cells of different cancer types and of different tissue origin possess a unique and characteristic profile of micro-RNAs (miRNAs) expression [77,78,79]. Naturally miRNAs epigenetically suppress translation of selected genes by binding and degrading complementary mRNA. Hirosawa et al. explored this quality of miRNA to design OFF switch by introducing miRNA-complementary sequence to the 5’ end of Cas9 mRNA. Therefore, in cells endogenously expressing selected miRNA (in this study the miR-21 specific to cancer or the miR-302 found in pluripotent stem cells) the Cas9 protein formation is inhibited and cleavage of targeted DNA does not occur. Researchers also tried creating ON switch, potentially more applicable in cancer gene editing therapy, using miRNA controlled expression of L7Ae protein and Cas9 mRNA preceded by L7Ae-binding motif. In this system the presence of miRNA blocks translation of L7Ae mRNA and, in the absence of this inhibiting protein, Cas9 is expressed without hindrance. Unfortunately, this model of miRNA-Cas9 ON switch proved to be inconsistent and did not entirely suppress CRISPR/Cas9 activity in non-cancerous cells [80].

Placing the expression of an AcrIIA4 protein under the control of miRNA upregulated in targeted tissue creates a loop similar to the one designed by Hirosawa et al., however, Acr inhibits Cas9 protein activity (as opposed to L7Ae which blocks expression of Cas9). As a result, this novel system shows no marked ’leakiness’ of Cas9 activity in untargeted cells and can be employed to various Cas9 orthologs, provided that there are Acrs capable of interacting with them [81,82,83,84]. The only downside to this system is the need for adjusting the quantity of delivered Acr plasmids to endogenous expression of selected miRNA, since an excess of Acr genes over miRNA would result in blocking Cas9 in targeted cells [85].

### 3.4. Expression Activated by Light

With the aim of developing an easily controlled and instantaneously modifiable expression activating signal, Qi et al., constructed a light-switch using Upstream activating sequence (UAS), Gal4 with an inserted vivid domain and P65 protein. Upon blue light irradiation, light-sensitive vivid induces Gal4 dimerization and binding to UAS, which leads to expression of downstream genes. In this study Cas13a was used as an effector with an sgRNA targeting the long non-coding RNA (lncRNA) of MALAT1 - oncogene in human bladder cancer. Cas13a is an alternative Cas protein with marked affinity to RNA molecules. Analogous to previously described light-controlled release of CRISPR/Cas9 from liposomes, Cas expression dependent on blue light might serve as a therapy for solid tumours after insertion of an optic fiber to cancerous tissue [84,86].

Blue light penetrates the skin only 1 mm deep, therefore the described system requires additional tools to exert therapeutic effect in internal organs and tissues. In contrast, Far-red light (FRL) reaches deeper than 5 mm into the skin and was used by Yu et al. to create a Cas9 switch. To reduce ’leakiness’ of the system, the Cas9 gene was split into two and the expression of the N-terminal fragment of Cas9 was placed under the control of FRL-sensitive FAST system composed of a photoreceptor, BphS and a transcriptional activator, p65-VP64-BldD. Upon irradiation with far-red light, both the constitutively expressed C-terminus and N-terminus would form a fully functional nuclease, with help of Coh2 and DocS binding proteins [87].

## 4. Targeted Modification of Cancer-Specific Sequences

### 4.1. Synthetic Lethality

Cancer specific genes are very rarely expressed exclusively in transformed cancerous tissue, thus altering them would also affect healthy cells of similar origin. An elegant solution may derive from the employment of synthetic lethality, a phenomenon described for various sets of genes, in which editing any one of these genes does not cause cell damage, but modifying both of them leads to cell death. If one of these paired genes is mutated during carcinogenesis and is present in all, or most of cancer cells, then genetic therapy targeted at the second gene would leave the healthy cells intact and affect only cells with pre-existing mutation of gene one [88,89,90]. So far, the CRISPR/Cas9 system was mostly used to screen for novel, druggable sets of genes inducing synthetic lethality, replacing the formerly prevailing methods based on RNA interference [88]. An alternative approach was used in a recent study conducted by Li et al. In this experiment sgRNAs targeting genes inducing synthetic lethality (CBP and p300) were placed under control of cancer-specific promoter (hTERT) and bladder-specific promoter (hUPII), the same promoters used in the logical AND gate described earlier [71]. As a result, dCas9-KRAB inhibited the expression of both of those genes only in bladder cancer cells, inducing their apoptosis [91].

### 4.2. Oncogenic Viruses

Opposed to the oncolytic viruses described previously as a potential therapeutic agent, oncogenic viruses exert a causal role in some types of cancer by inserting viral genes into the cellular genome, inducing tumorigenesis and enforcing sustained proliferation. Cancers resulting from these infections selectively posses and expres viral genes, making them an attractive target for specific genetic therapy [13]. Human papilloma virus (HPV), especially its most oncogenic types 16 and 18, is proved to cause cervical cancer in humans. The E6 and E7 genes introduced by HPV encode oncoproteins required for viability of infected cells that are responsible for their malignant transformation [92]. In multiple studies, the targeting of these viral genes (as well as the HPV-specific promoter) in cancer cells using the CRICPR/Cas9 system led to their apoptosis, while healthy cells were unaffected by this treatment [93,94,95]. A multitude of other oncogenic viruses have been described and treated using CRISPR/Cas9 gene editing including: Epstein-Barr virus (EBV) causing nasopharyngeal cancer, Burkitt’s lymphoma or Hodgkin’s lymphoma [96,97]; Hepatitis B and C viruses (HBV and HCV respectively) in hepatocellular carcinoma [98,99,100] and Human herpesvirus 8 (HHV-8) causing Kaposi’s sarcoma [101,102]. Many more neoplasms caused by oncogenic viral infections still await targeted genetic therapy, for example an adult T-cell leukemia/lymphoma induced by Human T- cell leukemia virus type 1 (HTLV-1) infection [103], or Merkel cell carcinoma - an aggressive skin cancer caused by Merkel cell polyomavirus (MCV) [104].

### 4.3. TERT Promoter Mutation

Telomerase activity is necessary for cancer cell immortalization. Certain point mutations in the promoter controlling telomerase gene (TERT) expression are implicated in many human cancers, including primary glioblastomas, creating novel binding sites for transcription factors and subsequently leading to increased promoter activity. Li et al. used Adenine base editor (ABE) fused with *Campylobacter jejuni* dead Cas9 (CjABE) to correct such G to A mutation located 124 base pairs upstream of the ATG start site for TERT gene. As mentioned, the CjCas9 gene is significantly shorter than SpCas9 enabling AAV-based delivery used in this study. CjABE was shown to successfully revert targeted oncogenic mutations, inducing senescence in cancerous cells, whereas cell lines with wild type TERT promoter sequences remained unaffected. Moreover, the therapeutic potential of CjABE targeting TERT promoter mutations was shown through the significantly inhibited tumour growth in athymic nude mice with intracranial glioblastoma cells. However, undesired modifications to bystander nucleotides within the editing window typical for base editors were also observed in this study, although they had no effect on TERT promoter activity [105].

## 5. Control over CRISPR/Cas9 Activity with Proteases

Oakes et al. successfully controlled the activity of the Cas9 protein by creating it in a circular permutation joined by an oligopeptide linker. At the same time, this interchangeable linker served as a sequence which was recognised by various viral proteases. A circular, bound form of Cas9 exerted no detectable nuclease activity, while the enzymatically degraded Cas9 protein regained its full potential for gene editing. Investigators have used it thus far to induce altruistic suicide in cells infected by certain plant or human viruses, but this mechanism to control Cas9 activity may be employed in the future either as a scaffold to create novel fusion proteins with functional domains or to target cells expressing endogenous, tissue-specific proteases [84,106].

## 6. CRISPR/Cas9 Modification of Non-Cancerous Cells Targeted at Defeating Cancer

The plethora of available options to target cancer cells with CRISPR/Cas9-based therapy offers great prospect into the future of oncological treatment, giving hope for better effectiveness and a reduced burden of adverse side effects. However, obstacles to the translation of cancer-selective genetic engineering first need to be addressed. The heterogeneity of a neoplastic cell population may mean that the gene, promoter or miRNA chosen as target of therapy (or its modifier) will not be upregulated or present in every cellular subpopulation. Therefore, some portion of cells could remain unaffected even after a successful and well-designed gene engineering, granting a growth advantage to the remaining cancerous cells and leading to a rapid re-population of a created niche [6,107]. Multiplexing, made possible by employing RNA-based recognition of DNA sequence selected for modification, could allow for targeting of broader set of tumour cells [108]. However, such strategies pose a great challenge to currently available delivery systems and still do not provide absolute certainty, that the entire cancerous population would be obliterated.

Alternatively to targeting cancer itself with genetic therapy, it might be more strategic and effective to modify stromal cells surrounding tumour tissue, that are more genetically stable and homogeneous. Their physiological function is to generally provide nourishment, protection and support for nearby functional cells. Unfortunately, they accommodate the growing cancer, adjusting to its needs. Modifying these pathologically changed stromal cells would deprive cancerous tissue of basic resources or shielding and likely result in remission. Moreover, disabling one metabolic pathway in cancer may be compensated by accompanying stromal cells, whereas disrupting the same crucial pathway in stromal cells in nutrient stress condition leads to death of cancerous cells [13,109].

Another approach at genetic therapies to treat malignancies is based on advances in immuno-oncology, by deploying the natural propensity of the immune system to destroy cancerous cells. Treatments currently focus on isolating T-cells from the patient, modifying their T-cell receptors (TCRs) or inserting CARs and re-introducing this anti-cancer lymphocyte population into the patients system [110]. Recently efforts have been made to prevent the exhaustion of the immune system that has been observed in the course of T-cell treatments, causing a diminishing effectiveness with time. One solution to cell exhaustion, which has proven effective in several studies, is to block expression of immune checkpoints regulators, like PD-1, CTLA-4 or LAG-3, on the surface of immune cells. These receptors described as crucial in cancer immunotherapy, often have ligands expressed on cancer cells, that deactivate the immune response [111,112,113,114]. Knocking-out of TGF-beta receptor II also reduced the exhaustion of CAR-T cells [115].

The necessity to create therapeutic cell populations for each patient individually prevents the scalability of immuno-oncological solutions. Moreover, the T-cell population in some patients suffering from an oncological disease is diluted and incapable of exerting therapeutic effect, especially if the patient has undergone radio- or chemotherapy. Therefore, it could be important to develop a universal CAR-T therapy based on allogenic T-cells from healthy donors, that would retain their activity and do not cause Graft-versus-host disease (GVHD). Efforts to these goals have already begun by depleting the CAR-T of the TCR and/or HLA molecules using CRISPR/Cas9 [39,116,117].

Beside immuno-oncology, the CRISPR/Cas9 system has been recently used ex vivo to modify cancerous tissue obtained from a murine host to develop a novel therapeutical agent. Cancer cells, reintroduced systemically to the animal of origin, possess the propensity to travel to sites of existing tumour and nest in their proximity. Reinshangen et al. recently explored this phenomenon to employ extracted and modified cells as a delivery vehicle for death signals (TRAIL) to original cancer. Cells were also equipped with a safety switch - a pro-drug converting enzyme HSV-TK, ensuring their sensitivity to ganciclovir administered after the eradication of original cancer cells [118].

## 7. Conclusions

CRISPR/Cas9 based gene editing provides a simple, cost-effective and precise method of manipulating the genome for the first time. As the gene editing toolbox continually expands with evolving technology, the potential to target genes implicated in cancer is immense. The central issue of using gene editing as a cancer therapy is the requirement of exceptionally high specificity. Sophisticated strategies to deliver this cancer-selectivity have been described employing targeted delivery, variable expression, the careful selection of targeted genes or utilizing cells naturally disposed to targeting cancer (Figure 1), with some of these already entering clinical trials (see Table 3). Although work relating to safety and efficacy remains, CRISPR/Cas9 based therapy could constitute a powerful weapon in the battle against cancer and create better outcomes for the patients of tomorrow.

## Figures and Tables

**Figure 1 ijms-23-00573-f001:**
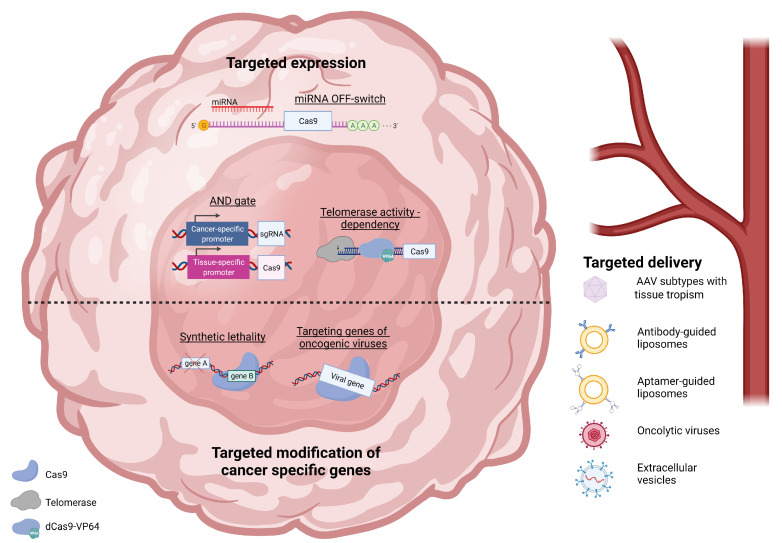
Summary of recently explored strategies for targeted CRISPR/Cas9 gene engineering in cancer therapy. **Targeted delivery** of CRISPR/Cas9 system can be achieved by employing vectors, such as: certain Adeno-associated virus serotypes with tropism to various tissues; liposomes guided to cancerous cells by specific antibodies or their protein-free counterparts - aptamers; oncolytic viruses replicating preferably in cancer cells or extracellular vesicles covered with guiding molecules derived from parent cell. **Targeted expression** in cancerous tissue can be based on: miRNA switch with cancer-specific miRNA causing degradation of Cas9 mRNA; AND gate created with promoters specific to cancer and tissue-of-origin, controlling the expression of different elements of CRISPR/Cas9 system; telomerase activity dependency with Cas9 protein being expressed only in tissues with high telomerase activity (i.e. highly proliferating). **Targeted modification of cancer-specific genes** can either focus on: synthetic lethality (cell death induced by disruption of certain pair of genes, where disruption of only one of those genes exerts no damage to the cell) with CRISPR/Cas9 system targeting one gene with the second one already mutated in cancerous cells; targeting genes derived from oncogenic viruses that drive the tumorigenesis and are necessary for survival of cancer. (Graph created with BioRender.com (accessed on 23 November 2021)).

**Table 2 ijms-23-00573-t002:** Pros and cons of most popular delivery methods [41,44].

Delivery Method		Pros	Cons
Capsule	Viral vector	Wide variety of available viruses; high effectiveness	Immunogenicity; pathogenicity; integration to genome
	AAV	Low immunogenicity; non-integrating; serotypes with tissue tropism	Low packaging capacity
	Lipid-based vector	Available surface modifications for targeted delivery	Low effectiveness; toxicity
Cas9 form	Plasmid	Stable; long lasting transfection	Slow generation of effects; requires delivery to nucleus
	mRNA	Effects observable faster; short-lived in cell (lower off-target activity)	Unstable; quickly degraded
	Protein	Editing immediately upon delivery	Difficult to deliver; contaminated with bacterial particles

**Table 3 ijms-23-00573-t003:** Examples of currently ongoing clinical trials, using CRISPR/Cas9 system for the treatment of cancer (found on ClinicalTrials.gov, acessed on 9 December 2021). [119,120,121,122,123].

Identifier	Study Title	Start Date	Phase	Disease/Condition
NCT05037669	Programmed Allogeneic CRISPR-edited T Cells Engineered to Express Anti-CD19 Chimeric Antigen Receptor (PACE CART19) in Patients With Relapsed Or Refractory CD19+ Leukemia and Lymphoma	January 2022	I	CD19+ Leukemia and Lymphoma
NCT03545815	Study of CRISPR-Cas9 Mediated PD-1 and TCR Gene-knocked Out Mesothelin- directed CAR-T Cells in Patients With Mesothelin Positive Multiple Solid Tumors	19 March 2018	I	Mesothelin Positive Solid Tumors
NCT04976218	TGF β R-KO CAR-EGFR T Cells in Advanced Biliary Tract Cancer	1 August 2021	I	Advanced Biliary Tract Cancer
NCT04426669	A Study of Metastatic Gastrointestinal Cancers Treated With Tumor Infiltrating Lymphocytes in Which the Gene Encoding the Intracellular Immune Checkpoint CISH Is Inhibited Using CRISPR Genetic Engineering	15 May 2020	I/II	Gastrointestinal Cancer
NCT03057912	A Safety and Efficacy Study of TALEN and CRISPR/Cas9 in the Treatment of HPV-related Cervical Intraepithelial NeoplasiaI	15 January 2018	I	Human Papillomavirus- Related Malignant Neoplasm

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
