# Peer review of "Targeting Cancer with CRISPR/Cas9-Based Therapy"

_ijms, 2022, doi:10.3390/ijms23010573_

Round 1
Reviewer 1 Report
The authors comprehensively summarize the current knowledge on the principles of precise CRISPR/Cas9-based therapeutic intervention as an alternative to conventional cancer therapy. Herein, they focus on the selective delivery and activity of CRISPR/Cas9 in order to eliminate malignant cells by targeting cancer-specific genes. They adopted an appropriate structure and produced a valuable review.
I would like to encourage the authors to add some details for prime editing to their ms (e.g., in line 79 and Table 1) as this is a very promising modification of the CRISPR/Cas9-sytem and considered an improvement of the described base editing (for an excellent review see doi.org/10.1038/s41587-020-0561-9).
In my opinion, the ms will also benefit from adding a paragraph or at least some information on the topic of undesired mutations occurring during CRISPR/Cas9-mediated editing. It is well known that genome editing techniques like CRISPR/Cas9 are limited by unintended mutations at off-target sites and at the targeted locus, respectively. This is of particular importance for therapeutic innervations and many efforts have been made to counteract this phenomenon (e.g., engineered high-fidelity versions of Cas9 to reduce off-target mutations and base/prime editors to reduce on-target mutations).
Specific points (in chronological order):
Line 4: consider changing the sentence to “discovery of the CRISPR/Cas9 system”
Line 10: consider changing the sentence to “based on the CRISPR/Cas9 system”
Line 11: consider changing the sentence to “delivery of the CRISPR/Cas9 machinery”
Line 45-46: the authors may want to make clearer that although one outcome of NHEJ are INDELs leading to disruption of the targeted sequence the result of this pathway is usually a scarless repair of the DSB. The current manuscript gives the impression that NHEJ always results in INDELs. In addition, these INDELs only lead to a KO if a frame-shift is induces in the coding region of the gene. The authors may want to rephrase the corresponding sentences accordingly.
Line 55-58: I suggest to rephrase the sentence as therapies with ZNF and TALENS are certainly possible but CRISPR-based therapies are simply more economic right now.
Line 62: I suggest to add this reference describing CRISPR as an adaptive defense mechanism of bacteria: doi.org/10.1126/science.1138140
Line 62-64: I am afraid I cannot follow the authors’ statement of “some molecular modifications”. The ms may benefit from explaining these modifications. Anyway, I am only aware of one modification, namely the fusion of crRNA and tracrRNA to one single-guide RNA (sgRNA).
Line 70: consider changing the sentence to “Cas9 protein, mediated by two nickase domains”
Line 77-79: I am afraid I have to disagree. To my knowledge base editors decrease the risk for undesired on-target mutations (like INDELs at the targeted locus) but not the risk for off-target mutations. On the contrary, they may even increase off-target mutations as they have been shown to be more promiscuous in targeting unspecific regions (doi.org/10.1038/s41587-020-0561-9).
Line 86: consider changing the sentence to “pyogenes (SpCas9)” and subsequently consider changing SpCas to SpCas9 (or Cas to Cas9) in the entire ms.
Table 1: The use of two Cas9 nickases resulting in a sticky end DSB is to my knowledge not considered to increase HDR but only to increase the specificity due to the need for consecutive action of two specific RNPs.
Line 144-145: consider changing the sentence to “than Cas9 found in Streptococcus pyogenes and Campulobacter jejuni Cas protein (CjCas9) gene”
Line 260-262: The authors may want to offer a hypothesis for this discrepancy of telomerase-driven CRISPR/Cas9 intervention between cancerous and non-cancerous cells with high telomerase activity
Line 344-352: I find it difficult to follow the authors’ conclusion in section 5. The ms may benefit from a more thorough explanation of the use of this system in cancer (I guess it is only useful in cancer derived from viruses which integrate into the genome?).
Figure 1.: Nice figure! But please export the figure from Biorender without branding (i.e., Created in BioRender.com)
In summary, upon minor revision the manuscript will add to the knowledge base and will be worth publishing.
Author Response
Authors’ Response to Reviewer’s comments
Journal: International Journal of Molecular Sciences
Title of Paper: Targeting Cancer with CRISPR/Cas9-Based Therapy
Authors: Katarzyna Balon, Adam Sheriff, Joanna Jacków, Łukasz Łaczmański
Submission date: 29 November 2021
Dear Reviewer,
Thank you for your time and effort spent on responding to the original version of our manuscript. All the issues pointed out in your comments have been addressed.
A short description underlining the versatility of prime editors, as an improvement to base editors, has been added. Additional paragraph on risk of off-target modifications has been written in the section “Introduction”, along with strategies to reduce the undesired mutations.
Apart from suggested changes, another paragraph was added during the course of revisions (subsection 4.3 TERT promoter mutation), describing the use of adenine base editor for correction of mutation activating TERT promoter in glioblastoma.
For detailed point-by-point response please see the attachment.
Yours faithfully,
Katarzyna Balon

Reviewer 2 Report
CRISPR/Cas9 editing is powerful tool and has shown great potential in cancer therapy, while this submission must do major revision before publication. (1) Many expressions should be more professional and precise. For example, the mainly targeting is oncogenes, the “Targeting Cancer” is precise? (2) CRISPR/Cas9 as drugs still has huge adverse effects, such as off-target and mutation, the authors should summary these points with more figures and tables; (3) the authors should introduce the preclinic and clinic trials of CRISPR/Cas9 in cancer gene therapies.
Author Response
Authors’ Response to Reviewer’s comments
Journal: International Journal of Molecular Sciences
Title of Paper: Targeting Cancer with CRISPR/Cas9-Based Therapy
Authors: Katarzyna Balon, Adam Sheriff, Joanna Jacków, Łukasz Łaczmański
Submission date: 29 November 2021
Dear Reviewer,
Thank you for your time and effort spent on responding to the original version of our manuscript and your excellent, accurate remarks. All the issues pointed out in your comments have been addressed.
Apart from suggested changes, another paragraph was added during the course of revisions (subsection 4.3 TERT promoter mutation), describing the use of adenine base editor for correction of mutation activating TERT promoter in glioblastoma. A short description underlining the versatility of prime editors, as an improvement to base editors, has also been written in the “Introduction”.
For a detailed point-by-point response please see attachment.
Yours faithfully,
Katarzyna Balon

Round 2
Reviewer 2 Report
It's ok for publishing now.